# Genome-Wide Association Study of Arabinoxylan Content from a 562 Hexaploid Wheat Collection

**DOI:** 10.3390/plants12010184

**Published:** 2023-01-01

**Authors:** Myoung Hui Lee, Jinhee Park, Kyeong-Hoon Kim, Kyeong-Min Kim, Chon-Sik Kang, Go Eun Lee, Jun Yong Choi, Jiyoung Shon, Jong-Min Ko, Changhyun Choi

**Affiliations:** Wheat Research Team, National Institute of Crop Science, Rural Development Administration, Wanju 55365, Republic of Korea

**Keywords:** wheat collection, genome-wide association studies, arabinoxylan content, whole wheat grain

## Abstract

The selection of wheat varieties with high arabinoxylan (AX) levels could effectively improve the daily consumption of dietary fiber. However, studies on the selection of markers for AX levels are scarce. This study analyzed AX levels in 562 wheat genotypes collected from 46 countries using a GWAS with the BLINK model in the GAPIT3. Wheat genotypes were classified into eight subpopulations that exhibited high genetic differentiation based on 31,926 SNP loci. Eight candidate genes were identified, among which those encoding F-box domain-containing proteins, disease resistance protein RPM1, and bZIP transcription factor 29 highly correlated with AX levels. The AX level was higher in the adenine allele than in the guanine alleles of these genes in the wheat collection. In addition, the AX level was approximately 10% higher in 3 adenine combinations than 2 guanine, 1 adenine, and 3 guanine combinations in genotypes of three genes (F-box domain-containing proteins, RPM1, and bZIP transcription factor 29). The adenine allele, present in 97.46% of AX-95086356 SNP, exhibited a high correlation with AX levels following classification by country. Notably, the East Asian wheat genotypes contain high adenine alleles in three genes. These results highlight the potential of these three SNPs to serve as selectable markers for high AX content.

## 1. Introduction

Dietary fiber (DF) is a vital component present in whole grains. It has health benefits, including lowering cholesterol and glucose levels and antioxidant activity [1,2,3,4]. Arabinoxylan (AX) constitutes the major source of DF in various grains, including wheat [5], and comprises a copolymer of two pentose sugars via a β-(1→4) linkage: arabinose and xylose. Some xylose residues are substituted by methyl-glucuronic acid and arabinose units, and arabinose can bind ferulate (FA) residues [6]. FA is the most abundant hydroxycinnamic acid derivative in plant cell walls [7], and it confers antioxidant properties to AX [8]. The dry weight content of the total dietary fiber in wheat is 9–20% [9,10,11] and consists of soluble and insoluble fractions. 

Wheat grain comprises three main components: the bran, germ, and endosperm. The outer layers are part of the bran, which is further subdivided into three distinct layers—the testa, aleurone, and pericarp [12]. Arabinoxylan is present in the starch endosperm, aleurone, and bran cell wall. A significant portion of AX from the wheat endosperm is present in the water extract fraction [13,14]. The AX of the aleurone layer and pericarp contains higher ferulic acid levels than that of starchy endosperm [15,16]. However, the structure and content of AX vary widely between genera, species, varieties, and tissues which affect the end-use properties and nutritional quality of the grain [17]. Variation in the AX content is related to differences in wheat processing and end-use quality [18,19,20]. The water-unextractable AX (WU-AX) fraction is generally considered to negatively impact gluten formation and baking quality. In contrast, the water-extractable AX (WE-AX) fraction affects moisture distribution between dough components and is generally associated with longer mixing times and larger loaf volumes [21].

The AX content and properties of wheat grains are influenced by the genotype and environmental conditions. A comparison of hard spring wheat and hard winter wheat suggested that the environment has a substantial effect on AX content relative to the genotype [22]. However, fungicide and nitrogen fertilizer rates do not affect WE-AX or total AX (TO-AX) content [23]. Additionally, the variation in AX content is mainly affected by the genotype [19,24,25,26]. Typically, the genotype and environment influence WE-AX and TO-AX levels. WE-AX content is primarily influenced by the genotype, whereas the WU-AX content within AX fractions is largely influenced by the environment [19,22,27,28].

Quantitative trait loci (QTL) associated with AX content were identified via genome-wide association studies (GWASs) in wheat and barley [29,30,31]. Nineteen QTLs encoding glycosyl hydrolase, glucosyltransferase, and glucan synthase genes were identified following analysis of the genetic variability of AX content from 104 tetraploid wheat genotypes [29]. Additionally, candidate genes for glycosyltransferases and glycosylases involved in AX biosynthesis or conversion associated with AX levels were identified in double-rowed barley [30]. Recently, single nucleotide polymorphisms (SNPs) were identified in predicted genes encoding glycosyl transferases, and these SNPs were translated into kompetitive allele-specific PCR (KASP) markers [31]. 

There is growing interest in the selection and development of wheat varieties with high AX levels owing to the benefits of whole grains. However, a comprehensive investigation is required to identify and select markers of AX levels. Therefore, the aim of our study was to perform a GWAS of the AX content from 562 hexaploid wheat samples harvested from 46 countries to identify SNPs. We also evaluated the AX content by allele and country of the allele. Finally, we aimed to identify the genomic regions and candidate genes associated with the AX content of wheat grains. We believe that the SNPs identified in this study will help develop KASP markers that will facilitate the selection and breeding of wheat varieties with a high AX content.

## 2. Results

### 2.1. Population Structure Analysis

A total of 562 wheat genotypes from 43 countries were analyzed to predict the effect of various genes on AX content using the Axiom 35 K iSelect SNP array. The kinship (VanRaden) matrix and structural parameters were calculated using Genome Association and Prediction Integrated Tool version 3 (GAPIT3) [32], and pairwise kinship matrices were depicted as heatmaps. Neighboring-joining genotype phylogenetic tree analysis revealed that the genotypes of 562 wheat genotypes collected from 46 countries represent eight major groups (I to VIII) with 183, 68, 72, 102, 38, 21, 77, and 1 genotype(s), respectively (Figure 1a). The pairwise kinship matrix heatmap of the 562 genotypes also revealed eight major groups, with their familial relationships displayed along the diagonal line with a few large blocks of closely related individuals (Figure 1b). Asian and American wheat genotypes accounted for 38% and 30% of the wheat core collection, respectively. This was followed by Europe, Eurasia, Africa, and Oceania, at 11%, 5%, 5%, and 1%, respectively. Unknown countries contributed 10% of the genotypes. A neighbor-joining tree indicated that the subgroups II, IV, and VI constitute 33.7% of the total studied population and have over 90% of genotypes in Korea. Groups I, III, V, VII, and VIII were genetically distant from the other groups. Groups V and VII had 45–70% of Mexican-derived genotypes. Groups I and III were a mix of genotypes from various countries. To better demonstrate the population structure of our study, we performed a principal component analysis (PCA) using the 562 wheat genotypes collected from 46 countries. When the genotypes were grouped by PCA based on continent, PC1 of the PCA biplot showed clear clusters of the Asian and American continental genotypes; however, the genotypes of four continental sets (Africa, Eurasia, Europe, and Oceania) were mostly mixed in the middle with no clear clusters (Figure 1c,d).

### 2.2. Arabinoxylan Content of 562 Bread Wheat Varieties 

The average AX level of all 562 wheat genotypes was 51 mg/g. The AX levels were compared for countries containing more than 10 genotypes (Appendix A, Figure 2). The AX level was highly variable for genotypes found in the same country, and the average AX level (56.7 mg/g) of the Republic of Korea genotypes (127) was higher than that of other countries. Japan contained the second highest variety of genotypes (10), with an AX content of 54.51 mg/g, similar to the Republic of Korea. In contrast, genotypes from China (31), Afghanistan (13), and Canada (10) had AX contents of 35.88 mg/g, 41.29 mg/g, and 42.11 mg/g, respectively, which were very relative to other countries. Genotypes from Ethiopia (22), Mexico (104), Russia (19), Turkey (12), and the USA (47) had AX contents of 45.75 mg/g, 48.56 mg/g, 50.03 mg/g, 44.95 mg/g, and 47.49 mg/g, respectively. In addition, there was high variability in the Afghanistan, China, and Mexico genotypes (Figure 2).

### 2.3. Genome-Wide Association Studies (GWASs) 

The GWAS was conducted for AX levels from wheat seeds harvested in 2019 using the Genome Association and Prediction Integrated Tool (GAPIT3) R package utilizing Bayesian information and the Linkage-disequilibrium Iteratively Nested Keyway (BLINK) model (Figure 3). GWAS results were visualized as Manhattan and quantile–quantile (Q-Q) plots [33]. Sixty-seven SNPs associated with the AX level trait were detected through a GWAS scan at a threshold of −log10(*p*) > 3.0. A −log10 (*p*-value) score of 5.0 was taken as the threshold for determining the linkage between a marker and the trait. Eight SNPs were detected through the GWAS scan to be associated with the AX level trait at a threshold of −log(*p*) > 5 on chromosomes 1BL, 3DL, 4BL, 5BL, 5DL, 7BS, and 7DS (Table 1). 

### 2.4. Candidate Gene Identification

Linkage disequilibrium (LD) was computed between 20 SNPs to further analyze the haplotype structure. The LD plot was constructed using the eight candidate SNPs (Appendix A). The LD heatmap of the SNPs on chromosomes 1B (AX–95086356), 3D (AX–94502724), 4B (AX–944703190, 5B (AX–95019636), 5D (AX–94534026, AX–34713015), 7B (AX–95092984), and 7D (AX–94934861) showed a low genetic correlation (0.0 to 1.0) between the SNPs near peaks adjacent to the putative gene for AX. The 20 adjacent SNPs ranged from 2.2–9.3 Mb in each genome (Appendix A). The six candidate SNPs are located on five different chromosomes. Two are located on the 5D chromosome but separated by 130 Mb.

Single nucleotide polymorphisms encode F-box-containing protein, disease resistance protein PIK6-NP-like, U2 small nuclear ribonucleoprotein A-like, serine/arginine repetitive matrix protein 1-like, bZIP transcription faction 29, carboxypeptidase, and disease resistance protein PIK6-NP-like on chromosomes 1B (1B–677224871), 4B (4B–609317748), 5D (5D–548367051, 5D–409878931), 7B (7B–67265891), and 7D (7D–2522044), respectively. No significant similarity was observed in the AX-94502724 probe sequence (Table 1).

The AX-95086356 SNP was located in the F-box/FBD/LRR-repeat protein at 685,782,123 bp out of a total of 700,547,350 bp in 1B. It is a nonsynonymous SNP in the first exon of a protein. The AX-94470319 probe was annotated with disease resistance proteins RPM1 and PIK6-NP-like proteins in CerealsDB and IWGSC RefSeq V2.1, respectively, and was located at 608,195,670 bp out of a total of 673,810,255 bp of 4B. The SNP of the AX-95019636 probe was located in the U2 small nuclear ribonucleoprotein A′-like protein located at 696,490,490 bp out of a total of 714,697,677 bp in 5B. The AX-94534026 SNP was located in the serine/arginine repetitive matrix protein 1-like protein at 548,367,051 bp of 569,951,140 bp in 5D. The AX-94713015 probe was located at 412,707,220 bp out of a total of 569,951,140 bp in 5D, which is the 3′UTR region of bZIP transcription factor 29. The SNP of the AX-95092984 probe was located in the carboxypeptidase gene located at 67,265,891 bp of 764,072,961 bp in 7B. Lastly, the AX-94934861 SNP was annotated with disease resistance proteins RPP13 and PIK6-NP-like proteins in CerealsDB and IWGSC RefSeq V2.1, respectively, and was located at 2,522,044 bp of 642,921,167 bp in 7D (Figure 4)

### 2.5. Arabinoxylan Distribution of the Genotypes for the Three SNPs 

The average AX levels from alleles of the SNPs in the wheat collection were compared to determine whether the eight selected SNP markers were associated with AX levels. The genotypes containing the SNP adenine genotype had an average AX grain content of approximately 5–9 mg/g higher than that containing guanine in three of the eight candidate SNPs: AX-95086356 (19.71%), AX-94470319 (17.71%), and AX-94713015 (10.48%) (Figure 5a–c, Appendix A). In addition, the genotypes containing the SNP cytosine or thymine genotype have an average AX content of 4–5 mg/g higher than the thymine or cytosine genotype in AX-94534026 (8.77%) and AX-95019636 (10.09%), respectively (Figure 5d,e, Appendix A). However, the AX levels by adenine/guanine alleles of AX-94934861 (7.2%) had less than 4 mg/g (Figure 5f, Appendix A). The thymine genotype and heterozygous (AC) genotype were absolute in the AX-94502724. The number of genotypes with adenine and guanin genotypes of AX-95086356 in the wheat collection was 214 and 91, respectively, and the average AX grain content was 53.08 mg/g and 44.74 mg/g, respectively. The number of genotypes with adenine and guanine genotypes of AX-94713015 in the wheat collection was 216 and 330, respectively, with average AX grain levels of 52.3 mg/g and 47.3 mg/g, respectively. In addition, the number of genotypes with adenine and guanine genotypes of AX-94470319 in the wheat collection was 174 and 383, respectively, with mean AX grain contents of 55.0 mg/g and 46.7 mg/g, respectively. Notably, the AX-95086356 genotype had 228 heterozygous (AG) genotypes similar to adenine homozygous genotypes, with an average AX grain content of 47.1 mg/g. Meanwhile, there were nine and three heterozygote genotypes in AX-94713015 and AX-94470319, respectively. However, the comparative analysis of AX grain content was uninformative owing to insufficient genotypes. The number of genotypes with adenine, cytosine, and heterozygous (AG) genotypes of AX-95092984 in the wheat collection were 10, 258, and 276, respectively, with an average AX content of 45.41 mg/g, 50.66 mg/g, and 48.24 mg/g, respectively. However, the average AX content of AX-95092984 was lower than that of the wheat collection. In addition, the average AX content of the remaining four SNPs (AX-94534026, AX-95019636, AX-94934861, and AX-94502724) was similar or lower than that of the wheat collection. These results show that each of the three SNP genotypes (AX-95086356, AX-94470319, and AX-94713015) contribute to differences in AX grain content. 

### 2.6. Arabinoxylan Content by Country According to the Three SNP Genotypes

The number of adenine genotypes excluding AG was 214 of 305, 216 of 546, and 174 of 558 genotypes, respectively, in AX-95086356, AX-94470319, and AX-94713015 accounting for 70.96%, 39.56%, and 31.18%. The AX levels were compared from countries with many genotypes, including Mexico, the USA, East Asia (China, Japan, Mongolia, North Korea, and the Republic of Korea), and the Republic of Korea (Figure 6a, Appendix A). The 61 genotypes in Mexico had 16, 21, and 24 adenine genotypes of AX-95086356, AX-94470319, and AX-94713015, accounting for 35.0%, 26.2%, and 39.3%, respectively. The 18 USA genotypes had 66.7%, 11.1%, and 27.8% adenine genotypes from 12, 2, and 5 genotypes, respectively. The 212 genotypes in East Asia had 97.46%, 74.78%, and 50.43% adenine genotypes from 115, 86, and 58 genotypes, respectively. Notably, the adenine genotype accounted for a much higher proportion than that in other countries in all three SNPs of East Asian genetic genotypes. The adenine genotype was absolute in the AX-95086356 SNP. In addition, 92, 77, and 48 out of 92 Republic of Korea genotypes accounted for 100%, 83.7%, and 52.2% of adenine genotypes, respectively. 

We compared AX contents by country (Mexico, the USA, East Asia, and the Republic of Korea) using three SNP genotypes. The AX content in adenine and guanine genotype SNPs of AX-95086356 were 58.62 mg/g and 45.06 mg/g in Mexico, 49.06 mg/g and 44.01 mg/g in the USA, 55.05 mg/g and 27.71 mg/g in East Asia, and 56.23 mg/g and 0 mg/g in the Republic of Korea, respectively (Figure 6b). The average AX content was higher in the adenine genotype than in the guanine genotype in Mexico (30.09%), the USA (12.45%), and East Asia (99.28%). However, Chinese and Republic of Korea genotypes had only three and no guanine genotype, respectively. The AX levels were 48.74 mg/g and 48.47 mg/g in Mexico and 50.85 mg/g and 46.94 mg/g in the USA for the AX-94470319 SNP when the genotypes were adenine and guanine, respectively, with lower than average AX content (Figure 6c). However, the USA genotype had only two adenine genotypes. The average AX levels were higher at 57.0 mg/g and 47.44 mg/g in East Asian genotypes and 57.39 mg/g and 50.31 mg/g in the Republic of Korea genotypes when the genotypes were adenine and guanine, respectively. The AX content was higher in the adenine genotype than in the guanine genotype in the USA (8.33%), East Asia (20.00%), and the Republic of Korea (14.07%). Similarly, the AX levels were 48.51 mg/g and 48.84 mg/g in Mexico and 48.74 mg/g and 46.84 mg/g in the USA in the SNP AX-94713015 when the genotypes were adenine and guanine, respectively, with lower than average AX content (Figure 6d). However, the AX levels in East Asia were 57.95 mg/g and 51.08 mg/g, and the AX levels in the Republic of Korea were 57.94 mg/g and 54.37 mg/g when the genotypes were adenine and guanine, respectively. AX content was higher in the adenine genotype than in the guanine genotype in the USA (3.97%), East Asia (13.04%), and the Republic of Korea (6.57%).

In AX-94534026 and AX-95019636, cytosine genotypes were 165 and 291 of 562 genotypes accounting for 29.26% and 51.60%, respectively, and thymine genotypes were 393 and 261 of 562 genotypes, accounting for 69.68% and 46.28%, respectively (Appendix A). The cytosine and thymine genotypes of SNPs from AX-94534026 were 51.56 mg/g and 44.83 mg/g in Mexico, 50.25 mg/g and 44.27 mg/g in the United States, 54.87 mg/g and 48.94 mg/g in East Asia, and 57.68 mg/g and 55.37 mg/g in the Republic of Korea, respectively (Appendix A). The AX levels in genotypes from Mexico, the USA, East Asia, and the Republic of Korea were 15.01%, 13.39%, 12.17%, and 4.17% higher in the cytosine genotype than in the thymine genotype. The AX content of AX-95019636 SNPs with cytosine and thymine genotypes was 45.53 mg/g and 52.11 mg/g in Mexico, 45.66 mg/g and 48.63 mg/g in the United States, 49.77 mg/g and 54.62 mg/g in East Asia, and 55.08 mg/g and 58.17 mg/g in the Republic of Korea, respectively (Appendix A). The AX levels in genotypes from Mexico, the USA, East Asia, and the Republic of Korea were 14.45%, 6.53%, 9.74%, and 5.63% higher in the thymine genotype than in the cytosine genotype. In AX-94934861, there were four, nine, and two adenine genotypes in the USA, East Asia, and the Republic of Korea, respectively (Appendix A). In AX-95092984, there were two, zero, four, and two adenine genotypes in Mexico, the USA, East Asia, and the Republic of Korea, respectively (Appendix A).

### 2.7. Arabinoxylan Content in Three Allelic Combinations

Next, we compared the average AX grain contents of the three selected SNP combinations (Figure 7). The number of genotypes of the adenine genotype combination of AX-94470319, AX-95086356, and AX-94713015 was 49, and the average AX grain content was 58.9 mg/g, which was the highest among the three SNP genotype combinations. Two SNP genotypes were adenine, and one SNP genotype was guanine, resulting in 98 genotypes with an average AX level of 54 mg/g. In a comparison of the AX levels between the two adenine-type combinations, the adenine–adenine–guanine combination in the order of AX-94470319, AX-95086356, and AX-94713015 scored 46 genotypes with an average AX value of 56 mg/g, and the number of genotypes with adenine–guanine–adenine and guanine–adenine–adenine were 5 and 47, respectively. The average AX levels were 54 mg/g and 52 mg/g, respectively. The average AX level of genotypes with the two adenine genotypes was 5 mg/g lower than that for all three adenine genotypes. The three SNP combinations (guanine–adenine–guanine, adenine–guanine–guanine, and guanine–guanine–adenine) with a single adenine genotype were 67, 32, and 18, respectively. In addition, AX levels were 49 mg/g, 44 mg/g, and 43 mg/g, respectively, and all were below 50 mg/g. The average AX level of the genotypes with the guanine genotype in all three SNPs was 45 mg/g; this was similar to genotype one adenine (Figure 7b). The AX content was higher in all three adenine genotypes (29.37%) and in the two adenine genotypes (18.52%) than in all guanine genotype.

### 2.8. Arabinoxylan Content of 41 Domestic Wheat Varieties

The AX content of 41 Korean genotypes bred and cultivated in the Republic of Korea was further analyzed (Appendix A). Among these genotypes, there is only one overlapping genotype (Keumkang), and the remaining 40 genotypes do not overlap with a 562 wheat collection. The AX content of the Republic of Korea genotypes was higher than that of other countries. The average AX content of 41 Republic of Korea varieties ranged from 41.11–67.58 mg/g, with an average of 58.1 mg/g. This was higher than the average of 56.91 mg/g from Republic of Korea cultivars of wheat collections (Figure 8a,b, Appendix A). Adenine genotypes were 100%, 51.22%, and 53.66% for AX-95086356, AX-94470319, and AX-94713015 SNPs, respectively, which correlates with the wheat collection (Figure 8c). However, the AX level in adenine and guanine genotypes of AX-94470319 were 58.11 mg/g and 58.09 mg/g, respectively, and the adenine and guanine genotypes of AX-94713015 were 58.39 mg/g and 57.27 mg/g, respectively (Figure 8d). In addition, the AX content was similar among the three SNP combinations of AX-95086356, AX-94470319, and AX-94713015 (Figure 8e).

The cytosine and thymine genotypes were 36.59% and 60.98% in AX-94534026, and 43.90% and 56.10% in AX-95019636, respectively. The AX level in cytosine and thymine genotypes for AX-94534026 was 59.63 mg/g and 57.19 mg/g, respectively. The AX level in cytosine and thymine genotypes for AX-95019636 was 57.63 mg/g and 58.46 mg/g, respectively. The average AX content was similar in AX-94534026 and AX-95019636 alleles (Appendix A).

## 3. Discussion

Grain fiber has significant benefits for human health, leading to the growing use of whole grains and an increase in the recommended daily dietary fiber intake for maintaining a healthy lifestyle. However, despite the benefits of fiber intake, fiber intake is still below recommended levels. In this study, we used 562 hexaploid wheat varieties to identify significant genomic regions associated with AX content variation in whole grains. The genotypes of 562 wheat genotypes collected from 46 countries represent eight major groups. However, the collected wheat genotypes are accounted for in a specific country. In particular, the Republic of Korea (22%) and Mexico (18%) account for a larger share compared with Eurasia (5%), Africa (5%), and Oceania (1%). Therefore, determining the trend of AX content by country was challenging. Notably, subgroups II, IV, and VI have over 90% of Republic of Korea genotypes and were genetically distant from other groups in the neighbor-joining tree. 

The AX content of wheat grains is affected by genotype and environmental conditions. The WE-AX content is mainly affected by the genotype, while the WU-AX content is more so influenced by the environment [16,19,24,25]. In the present study, the AX content of whole grains was high in Korean and Japanese wheat genotypes, but low in those from other Asian countries, and there was a large difference in AX content between genotypes. These results indicate the strong effect of genotype on AX levels in whole grains from 562 wheat genotypes, which were cultivated and harvested under the same conditions.

We identified three candidate genes that were not previously identified using GWAS analysis. AX-95086356 is present in the first exon of the F-box/FBD/LRR-repeat protein. The gene function of the wheat F-box/FBD/LRR-repeat protein was not previously reported, and its ortholog (Arabidopsis At2g04230) was limited to expression analysis [34]. It likely has the characteristics of a disease-resistance-related protein since it contains a leucine-rich repeat (LRR) domain. BLAST results for the AX-94470319 probe were annotated with disease resistance protein RPM1 in the CerealsDB. RPM1 is a nucleotide-binding site (NBS)-LRR protein that is required for resistance to *Pseudomonas syringae* expressing type III effectors (AvrRpm1 or AvrB) in *Arabidopsis thaliana* [35,36]. The *TaRPM1* gene in wheat is upregulated approximately six times following infection by *Puccinia striiformis* f. sp. *Tritici* (*Pst*) under high temperatures compared with that under normal temperatures [37]. The *Oryza sativa* RPM 1-like resistance gene 1 (OsRLR1) is a putative homologous gene of RPM 1 that is involved in the disease resistance pathway to fungi (*Pyricularia oryzae*) and the bacterial pathogen (*Xanthomonas oryzae* pv. *Oryzae*) causing rice blast and blight diseases, respectively [38]. Finally, AX-94713015 SNP is located in the 3′UTR (240 bp downstream from the stop codon) of the basic leucine zipper (*bZIP*) transcription factor 29. Therefore, it is unlikely to affect the function of this gene. The bZIP transcription factor family is a large transcriptional regulatory family involved in abiotic or biotic stress in plants and exhibits a wide range of functions [39,40]. A total of 227 TabZIP gene family members were determined from the wheat genome database [41], and they have diverse biological functions [42,43,44,45,46,47,48]. However, distinguishing the sequence of probe AX-94713015 from many bZIP genes is challenging. Alteration of cell wall polysaccharides significantly affects plant resistance to various pathogens because it leads to the activation of defense signaling pathways [49,50]. Therefore, the change in the disease resistance gene according to AX content is plausible. 

The selection of DF-enriched whole-grain varieties is likely the most effective way to increase DF intake. Three SNP genotypes (AX-95086356, AX-94470319, and AX-94713015) had an average AX grain content that was approximately 5–9 mg/g higher in the adenine genotype than in the guanine genotype. Among these, the average AX content was higher in the adenine genotype than in the guanine genotype of AX-95086356 from Mexico, the USA, and East Asia. However, the average AX content was similar in adenine and guanine alleles in Mexico and the USA; there were differences only in East Asia and the Republic of Korea in both AX-94470319 and AX-94713015 SNPs. We propose that these results are because of the low average AX content in Mexico and the USA. The AX content is variable, and low on average from the 31 Chinese genotypes. The estimated average increase in AX content in East Asia is owing to the relatively large number of genotypes from Republic of Korea genotypes (127).

The average AX content in the three adenine genotype combinations was slightly higher than in each genotype and higher than in the three guanine genotype combinations. However, there was no difference in AX content among the 41 domestic varieties according to SNPs. This is probably owing to the selection and breeding of genetically similar cultivars, and the average AX content of 41 Korean cultivars is estimated to be similarly high. These results suggest that the SNPs found in this study have limitations in the selection of varieties with a high AX content among genotypes with a high or low AX content. However, three SNPs were associated with variations in AX content and were highly correlated with AX content according to the SNP allele of AX-95086356 and the three SNP combinations. It is expected that wheat varieties with a high AX content can be selected in breeding programs through the translation of kompetitive allele-specific PCR (KASP) markers using the most vital 1BL chromosome SNP identified through GWAS analysis.

## 4. Materials and Methods

### 4.1. Materials and Arabinoxylan Extraction

We constructed a collection of wheat genotypes from 614 out of 1967 wheat lines [51]. The 562 genotypes analyzed in this study included some of the genotypes used by Yang et al. (2021) and 562 hexaploid wheat genotypes collected from 46 countries [31]. This included 127 from the Republic of Korea, 104 from Mexico, 47 from the USA, 31 from China, 22 from Ethiopia, 19 from Russia, 13 from Afghanistan, 10 from Canada, and 10 from Japan. The 57 genotypes were of unknown origin. Additionally, 41 Republic of Korea wheat cultivars were included in this study (Appendix A). All lines were grown in the fields at the National Institute of Crop Science (Wanju, Republic of Korea) during the 2018–2019 cropping cycles. The AX content values of whole grains used in this work were extracted from a previous study [52]. 

### 4.2. DNA Extraction and Genotyping

Genomic DNA was extracted from the young leaves of 10-day-old seedlings grown on Petri dishes with moisture paper. Genomic DNA was extracted from leaf tissue using the Higene^TM^ Genomic DNA prep kit (solution type; BioFACT, Republic of Korea). A panel of 562 wheat accessions was genotyped using the 35 K SNP iSelect chip array (DNA link Co., Seoul, Republic of Korea) containing 35,143 SNP markers [53].

### 4.3. Genome-Wide Association Analysis and Population Structure 

Genotyping data were filtered by removing 3271 markers with no chromosomal position in the IWGSC v1.0 genome sequences [54]. Thus, 31,926 polymorphic SNP markers were used for subsequent analyses (DNAcare Co., Seoul, Republic of Korea). Bayesian information and Linkage-disequilibrium Iteratively Nested Keyway (BLINK) [55] were implemented in the Genome Association and Prediction Integrated Tool version 3 (GAPIT3). R package was used to compute associations between the SNPs and traits [32]. The R package ”qq-man” was used to visualize the GWAS results as Manhattan and quantile–quantile (Q-Q) plots [54]. Principal component analysis (PCA) and kinship (VanRaden) matrices were calculated using GAPIT3. The population structure was assessed using PCA, and the percentage of variation explained by the first two principal components was obtained. LD analyses were performed for significant SNPs using QTLmax 3.0 (Katy, Texas, USA) [56]. 

### 4.4. Identification of Putative Candidate Genes

The probe sequences of the significant SNP markers were identified in the Ensembl Plants database and were searched against the reference sequence IWGSC RefSeq V2.1 [57] in the National Center for Biotechnology Information (NCBI) using the BLAST tool. The nucleotide sequence of IWGSC RefSeq V2.1 was updated by approximately 10% compared to the RefSeq V1.0 [57]. Next, a schematic representation of the physical map of hexaploid wheat was developed with significant SNPs associated with AX content. 

## 5. Conclusions

A collection of 562 wheat genotypes was analyzed for the AX content of whole wheat grains, and GWAS analysis was conducted to identify genomic regions associated with AX content. East Asian wheat genotypes had a higher ratio of adenine in the three SNPs than in wheat accessions from other countries; in particular, the ratio of adenine in the SNPs located in 1BL was substantially high. The combination of the 1BL, 4BL, and 5DL genomic regions had an effect on AX accumulation. Among these, the 1BL genomic region had the greatest influence on AX accumulation. These results suggest that several loci located in the 1B, 4B, and 5D genomes are involved in determining the AX content, and the translation of KASP markers is expected to facilitate the selection of wheat varieties with a high AX content in breeding programs.

## Figures and Tables

**Figure 1 plants-12-00184-f001:**
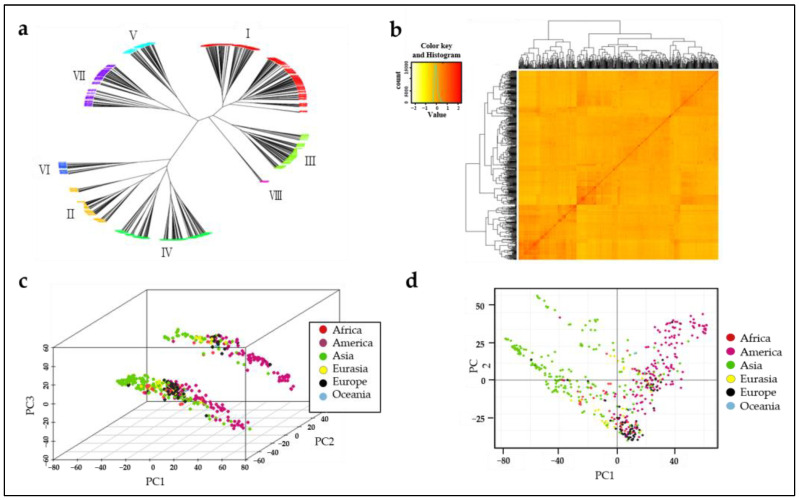
Population structure of the 562 wheat genotype varieties. Kinship is virtually displayed via a neighbor-joining tree (**a**) and a heatmap (**b**). These principal components (PC) are virtually displayed in three dimensions (**c**) and two dimensions (**d**). The different colors represent the six populations.

**Figure 2 plants-12-00184-f002:**
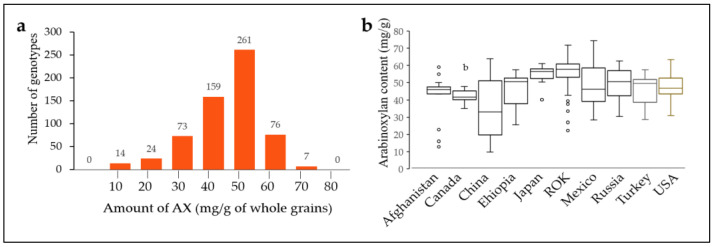
Arabinoxylan (AX) level evaluated in a collection of 562 *Triticum aestivum* genotypes. AX level distribution diagram (**a**) and AX level by country (**b**). ROK, Republic of Korea.

**Figure 3 plants-12-00184-f003:**
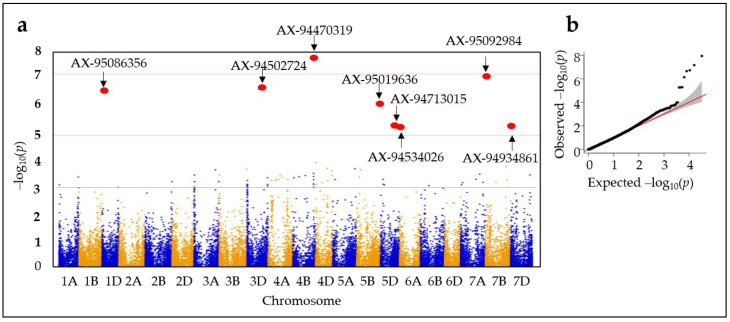
Manhattan (**a**) and quantile–quantile plots (**b**) of genome-wide associations study (GWAS) results of AX levels from 562 wheat genotypes. Eight SNPs associated with AX contents at −log(*p*) > 5 thresholds through GWAS analysis are indicated by arrows.

**Figure 4 plants-12-00184-f004:**
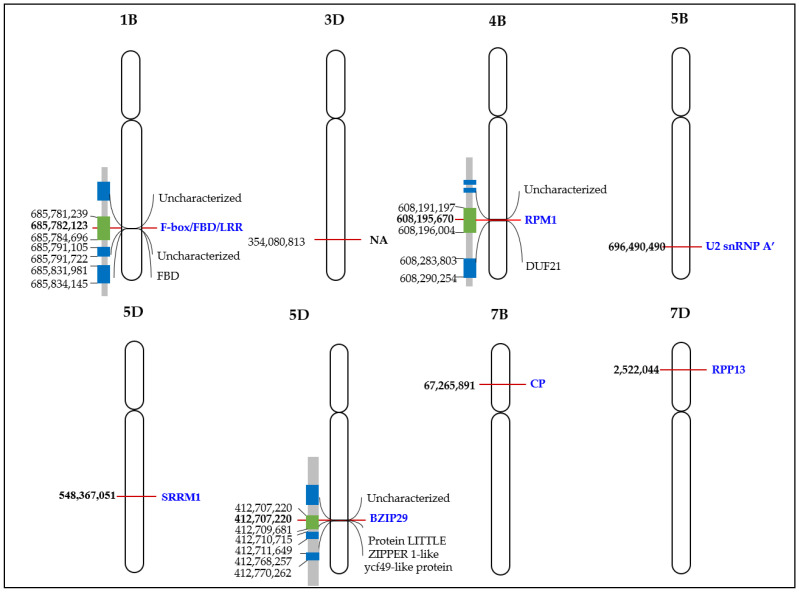
Physical map of the location of eight SNPs and related gene *Triticum aestivum* associated with AX level. The left and right side show the position and SNP for each locus, respectively. Candidate genes are named in blue. *F-box/FBD/LRR* = *F-box/FBD/LRR-repeat protein; FBD* = *FBD-associated F-box protein*; *RPM1 = disease resistance protein RPM1; DUF21* = *DUF21 domain-containing protein; U2 snRNP A′* = *U2 small nuclear ribonucleoprotein A′-like; SRRM1* = *serine/arginine repetitive matrix protein 1-like; BZIP29* = *bZIP transcription factor 29-like; CP* = *carboxypeptidase; RPP13* = *disease resistance protein RPP13*.

**Figure 5 plants-12-00184-f005:**
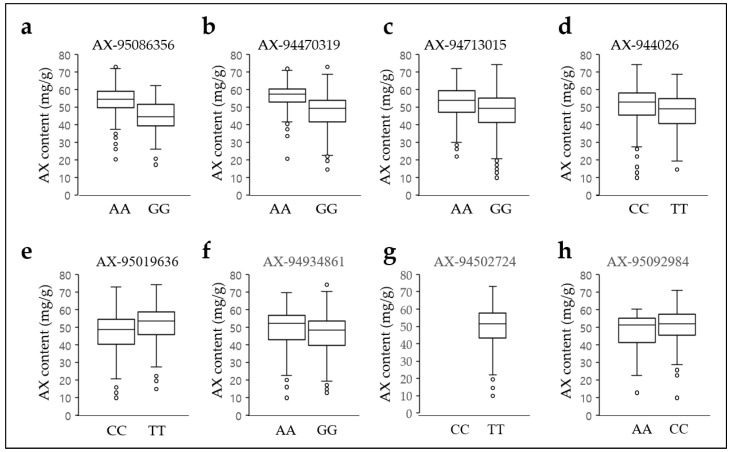
Arabinoxylan content of each allele in the eight SNPs. Arabinoxylan level in allele AX-95086356 (**a**), AX-94470319 (**b**), AX-94713015 (**c**), AX_94534026 (**d**), AX-95019636 (**e**), AX-94934861 (**f**), AX-94502724 (**g**), and AX-95092984 (**h**). The X-axis depicts the two alleles, and the Y-axis represents the AX content. AX, arabinoxylan; A, adenine; C, cytosine; T, thymine; G, guanine.

**Figure 6 plants-12-00184-f006:**
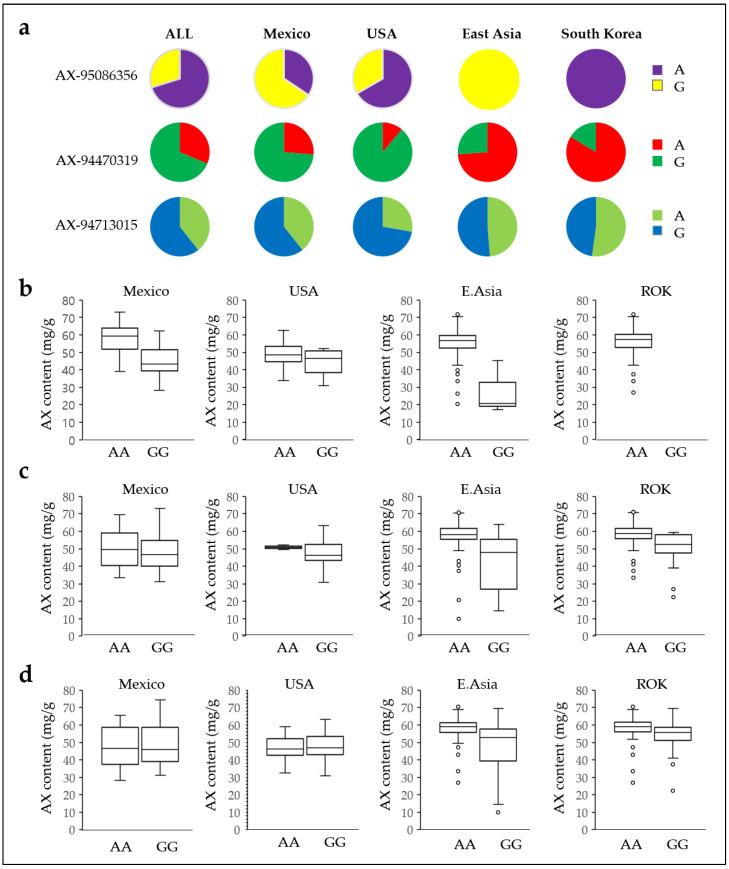
Arabinoxylan level by country according to the three SNP alleles. (**a**) Adenine and guanine rates by country; (**b**) AX level by country according to the SNP allele in AX-95086356; (**c**) AX-94470319; and (**d**) AX-94713015. The X-axis depicts the two alleles, and the Y-axis represents the AX content. AX, arabinoxylan; A, adenine; G, guanine. ROK, Republic of Korea.

**Figure 7 plants-12-00184-f007:**
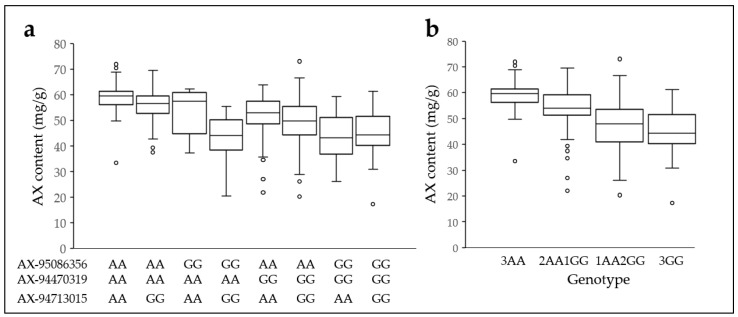
Arabinoxylan distribution of three allele combinations. Arabinoxylan content in eight combinations of three SNPs (**a**) and three adenine, two adenine and one guanine, one adenine and two guanine, and three guanine combinations (**b**). AX, arabinoxylan; A, adenine; G, guanine.

**Figure 8 plants-12-00184-f008:**
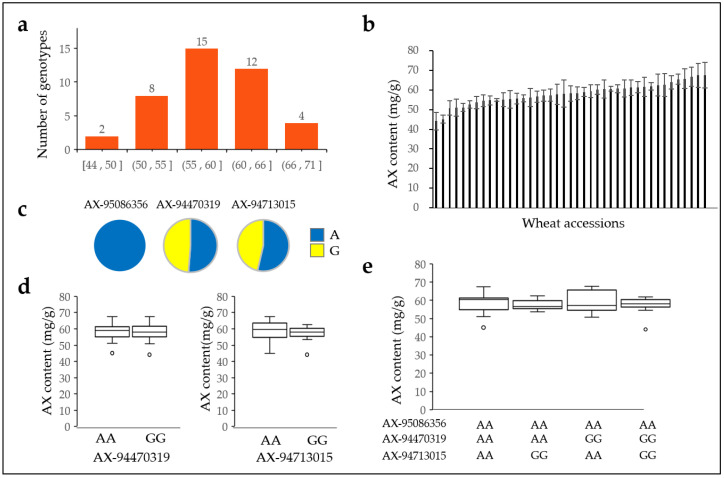
The AX content in 41 Republic of Korea cultivars. Distribution of AX content in 41 Republic of Korea cultivars (**a**,**b**), adenine and guanine allele rates in AX–95086356, AX–9447319, and AX–94713015 (**c**), AX content in AX-94470319 and AX-94713015 alleles (**d**), and combination of AX–95086356, AX-94470319, and AX-94713015 alleles (**e**). The number on the bar indicates the number of genotypes.

**Table 1 plants-12-00184-t001:** Candidate genes and their functions for each significant single nucleotide polymorphism (SNP) associated with arabinoxylan levels.

SNP	Chr.	Position	*p*-Value	RefSeq Sequence	FDR	Gene Name	Alleles
AX–95086356	1BL	677224871	2.24 × 10^−7^	NC_057795.1	0.000791	F-box/FBD/LRR-repeat protein	A/G
AX–94502724	3DL	354080813	1.90 × 10^−8^	NC_057802.1	0.001791	NA	C/T
AX–94470319	4BL	609317748	1.15 × 10^−8^	NC_057804.1	0.000369	Disease resistance protein RPM1/PIK6-NP-like	A/G
AX–95019636	5BL	696490490	7.41 × 10^−7^	NC_057807.1	0.00473	U2 small nuclear ribonucleoprotein A′-like	C/T
AX–94534026	5DL	548367051	5.47 × 10^−6^	NC_057808.1	0.021813	serine/arginine repetitive matrix protein 1-like	C/T
AX–94713015	5DL	409878931	5.27 × 10^−6^	NC_057808.1	0.021813	bZIP transcription factior 29	A/G
AX–95092984	7BS	67265891	6.94 × 10^−8^	NC_057813.1	0.001109	Carboxypeptidase	A/C
AX–94934861	7DS	2522044	5.39 × 10^−6^	NC_057814.1	0.021813	Disease resistance protein RPP13/PIK6-NP-like	A/G

## Data Availability

Not applicable.

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
