# Peer review of "Genome-Wide Association Study of Arabinoxylan Content from a 562 Hexaploid Wheat Collection"

_plants, 2023, doi:10.3390/plants12010184_

Round 1

Reviewer 1 Report

This manuscript reports on 1) the analysis of the population structure of a bread wheat collection, comprising accessions from 47 countries, based on genome-wide distributed SNP markers; 2) the analysis of grain arabinoxylan content of the same population; 3) the identification of eight significant SNP markers associated with the grain arabinoxylan content by GWAS analysis.

The topic merits research attention as arabinoxylans are the major components of the cereal dietary fibers and are recognized for their health-promoting effects. So, studies on the genetics of arabinoxylan content in bread wheat as a staple cereal could provide beneficial knowledge for the molecular breeding of genotypes of higher fiber content.

The accent of this work is on three putative candidate genes hit directly by the significant SNPs, followed by very detailed comparative analysis of genotypes carrying different alleles at the three SNP loci with respect to arabinoxylan content.  

My major concerns about this manuscript are the following:

First, the search for candidate genes is restricted to the genes that are directly hit by the significant SNPs. The authors do not explain the putative relation of the products of these genes with the trait under investigation. My opinion is that useful information could have been obtained if the authors had extended their search based on linkage disequilibrium (LD). However, they do not report on analyzing the genetic diversity and LD in the studied population. 

Second, the authors claim significant differences between genotypes carrying different alleles at three SNP loci (Fig. 5 and Fig 6) denoted by asterisks. Looking at the bars on the columns (which, I suppose, denote the standard deviations, though it is not mentioned explicitly), I would say that these differences are definitely not statistically significant. Based on the lack of significant differences between the sets carrying different alleles, I do not see any sense of further discussing these findings.

Third, the plant material used in this study is not clarified enough. What is the number of studied accessions – 567 or 562 or else? Authors refer to Yang et al 2021 [31] for information, but I was not able to check this reference, since it is in Korean. Neither I was able to deduce the number of studied accessions from the numbers mentioned on page 10, lines 346-350.

Fourth, authors are strongly advised to be very careful when using pivotal terms. Thus, at 13 places within the text they used “cysteine” (which is an amino acid) instead of “cytosine”. Also, they used “guanidine” at 5 places instead of “guanine”.

Author Response

Reviewer 1

This manuscript reports on 1) the analysis of the population structure of a bread wheat collection, comprising accessions from 47 countries, based on genome-wide distributed SNP markers; 2) the analysis of grain arabinoxylan content of the same population; 3) the identification of eight significant SNP markers associated with the grain arabinoxylan content by GWAS analysis.

The topic merits research attention as arabinoxylans are the major components of the cereal dietary fibers and are recognized for their health-promoting effects. So, studies on the genetics of arabinoxylan content in bread wheat as a staple cereal could provide beneficial knowledge for the molecular breeding of genotypes of higher fiber content.

The accent of this work is on three putative candidate genes hit directly by the significant SNPs, followed by very detailed comparative analysis of genotypes carrying different alleles at the three SNP loci with respect to arabinoxylan content.  

My major concerns about this manuscript are the following:

First, the search for candidate genes is restricted to the genes that are directly hit by the significant SNPs. The authors do not explain the putative relation of the products of these genes with the trait under investigation. My opinion is that useful information could have been obtained if the authors had extended their search based on linkage disequilibrium (LD). However, they do not report on analyzing the genetic diversity and LD in the studied population. 

Response:

Thanks for your comments.

We analyzed linkage disequilibrium (LD) and explained in main text.

Second, the authors claim significant differences between genotypes carrying different alleles at three SNP loci (Fig. 5 and Fig 6) denoted by asterisks. Looking at the bars on the columns (which, I suppose, denote the standard deviations, though it is not mentioned explicitly), I would say that these differences are definitely not statistically significant. Based on the lack of significant differences between the sets carrying different alleles, I do not see any sense of further discussing these findings.

Response:

Thanks for your opinion.

The three alleles of AX-5086356, AX-94470319 and AX-94713015 showed statistically significant differences in Fig 5. In addition, AX content did not show a significant difference in these three alleles in the United States and Mexico, but showed a significant difference in East Asia and Korea. And, as shown in Fig 7, since the three adenine combinations showed a statistically higher AX content than the three guanine combinations, these results are considered to be significant alleles for the arabinoxylan content.

We have included ANOVA tables of AX with df, p-values and significance level for genotype and country in Supplementary File 1.

Third, the plant material used in this study is not clarified enough. What is the number of studied accessions – 567 or 562 or else?

Response:

562 genotypes were analyzed. We corrected.

Authors refer to Yang et al 2021 [31] for information, but I was not able to check this reference, since it is in Korean. Neither I was able to deduce the number of studied accessions from the numbers mentioned on page 10,ines 346-350.

Response:

We included information of 562 wheat genotypes and AX content in Supplementary Table S1 and Supplementary File 1.

Fourth, authors are strongly advised to be very careful when using pivotal terms. Thus, at 13 places within the text they used “cysteine” (which is an amino acid) instead of “cytosine”. Also, they used “guanidine” at 5 places instead of “guanine”.

Response:

Thanks for your comments.

It’s corrected.

Reviewer 2 Report

The manuscript "Genome-wide association study of arabinoxylan content from 567 hexaploid wheat collections" deals with an interesting topic and the presented results would be valuable for readers of the journal.

I would recommend it for publication after its revision.    

Some parts of the manuscripts lack clarity and precision in terminology, which hinders their comprehension. This is not very problematic, but some additional proof reading and English editing would surely improve the clarity and style of the paper.  

Title

Are there 567 collections of hexaploid wheat or one collection consisted of 567 hexaploid wheat genotypes? The title implies the former, but if the latter is true, please correct the title to „Genome-wide association study of arabinoxylan content from а 567 hexaploid wheat collection”.

Abstract

Line 14: There are two words „using” in a sentence. Replace one of them with “with”.

Lines 19-20: “In addition, the AX level was approximately 10% higher in 3 adenine combinations than 2 guanine, 1 adenine, and 3 guanine combinations in genotypes of three genes.” Please write the names of these genes either here or in the last sentence of the abstract (Line 23).

Combine in one sentence “The adenine allele of AX-95086356 exhibited a high correlation with AX levels following classification by country” (Lines 20-21) and “and the AX-95086356 SNP has 97.46% adenine alleles” (Lines 22-23), e.g. “The adenine allele, present in 97.46% in AX-95086356 SNP, exhibited a high correlation with AX levels following classification by country.”

Line 22-23: Please, write the genotype name after “the East Asian wheat genotype” and remove “and the AX-95086356 SNP has 97.46% adenine alleles.”

Introduction

Lines 33-34: “Some xylose residues are substituted by methyl-glucuronic acid and arabinose units, and arabinose can bind ferulate (FA) residues [6]”. What does this mean in practice? Do such substitutions bring some different properties to DF and how it is relevant to this study? Please add in a very few words what this practically implies in the context of your study.

Line 36: “Arabinoxylan is present in the endosperm of bran, aleurone, and bran cell wall”. This sentence is not very clear. It may imply that there is “the endosperm of bran”. (Bran, endosperm and germ are three different distinctive parts of wheat grain). This sentence needs to be unambiguous and precise. It may be also useful to briefly and precisely remand the readers of the parts of the grain, i.e. in which part of the grain aleurone and pericarp are located, and keep consistent in the text.

Line 38-39: “The AX of the aleurone layer and pericarp contains higher ferulic acid levels than that of starchy endosperm [12, 13].” Similar to my previous comment in Lines 33-34, the Introduction part should introduce the readers to the meaning and importance of the research, so please add very briefly to this sentence why higher ferulic acid levels are important (e.g. antioxidant properties, etc.).

Line 66-72: Replace the “Therefore, we performed a” with “Therefore, the aim of our study was to performed a” and rewrite the rest of the paragraph in such style. This should not be the part where you present your results (there is a Results section for that purpose), but the emphasis what is the aim of your study. The last sentence where the authors express their believes should be move to the conclusion.

Results

Lines 75-76: “Recently, we constructed a core collection of wheat resources from 614 out of 1,967 wheat lines [29].” It is not clear whether or not this core collection or a part of it is used in this study. If this material is used in this study, this sentence should be moved to the Material and methods section and elaborated. If not, the authors could compare and discuss their results with the finding in the reference no. 29 in the Discussion section.

Line 76: “A total of 567 wheat genotypes were obtained using the Axiom 35 K iSelect SNP array”. Were 567 or 562 genotypes analyzed? 

Line 81: How 567 (the sum of  184, 68, 72, 102, 38, 21, 77, and 5) genotypes were analyzed and clustered when it is stated in the Material and Methods that 5 out of 567 genotypes were excluded from genotyping?

Line 85: The “main contributions” to what? This is not precise. Did you mean that 38% and 30% of all wheat genotypes were from in Asia and America, respectively? This has to be written more clearly.

Line 91: What does this sentence mean? “Groups I and III genetic traits were derived from various countries.” Please, rewrite it and be clearer.

Line 92: This sentence should be in MM section.

Lines 93-94: This statement has no meaning. The author should describe how PCA divided/grouped/ classified the genotypes.  

Lines 100-101: The Results section should contain only the results of this study and should not have any citations. This sentence belongs to MM.  

I would like to see the full ANOVA table of the AX with df, p values and levels of significance for the genotypes and countries, as the source of variation.

Figure 2a. Replace “germplasm” with “genotypes”.

Figure 2b. It would be useful to see the letters above the bars to see whether or not there are significant differences among the average values of AX in different countries. Why only 11 countries are presented in the figure? From how many countries were the genotypes anyway? The authors discussed (Lines 302-304) that some groups were underrepresented (with few accessions) and thus not presented, but this should be also appropriately stated in the results, just before showing the results, so readers can follow.   

Lines 113-115: This is a part of the Material and methods section. It should be move there.

Lines 136-137, 138: Cite the databases in MM. The same is for [34].

Figure 6 Word “East” is missing on a graph.  

Figure 8a. Replace “germplasm” with “genotypes”.

Discussion

Discussion is rather short and should be elaborated a little bit more. For example, what analysed genotypes could be potentially valuable as a source of AX for wheat improvement. Discuss the difference between the level of AX in genotypes from South Korea (and e.g. Japan) and from other countries. How could be explained. What are the previous findings on AX content?

Material and methods

Lines 346-347: “The resources analyzed in this study were a subset of the resources used by Yang et al. (2021) and 567 hexaploid wheat varieties collected from 47 countries [31]”. Having stated 567 hexaploid wheat varieties in the second part of the sentence, what did the authors mean by “a subset of the resources”? The statement suggests that there is a subset of some resources AND 567 varieties. The cited paper [31] with the information about the varieties and AX context that the authors used for this study is in Korean language. Please, add the remaining genotypes, the countries and their AX values to the 41 varieties in the supplementary file. All 562 genotypes with their data must be included in the supplementary file.

Lines 89, 102, 104, 109, 164, 172, 174, 177, 180, 181, 183, 196, 197, 203, 234, 274, 294, 235, 346, 359 (and probably some more): The term “resources” in this context is a bit odd and inappropriate. Please, use more suitable terms, such as genotypes, accessions or varieties.

Lines 358-359: If out of 567 accessions, 562 were genotypes and 5 were excluded from the analyses due to abnormal phenotype, it cannot be claimed that all 567 accessions were analyzed. It is 562 then. This must be corrected in the whole manuscript.

Conclusion

Lines 388: Correct “three SNPs than in other countries” to “three SNPs than in wheat accessions from other countries”. The wheat varieties can be in/from other countries, not the SNPs. 

Author Response

Reviewer 2

The manuscript "Genome-wide association study of arabinoxylan content from 567 hexaploid wheat collections" deals with an interesting topic and the presented results would be valuable for readers of the journal.

I would recommend it for publication after its revision.    

Some parts of the manuscripts lack clarity and precision in terminology, which hinders their comprehension. This is not very problematic, but some additional proof reading and English editing would surely improve the clarity and style of the paper.  

Title

Are there 567 collections of hexaploid wheat or one collection consisted of 567 hexaploid wheat genotypes? The title implies the former, but if the latter is true, please correct the title to „Genome-wide association study of arabinoxylan content from а 567 hexaploid wheat collection”.

Response:

Thanks for your comments.

We corrected the title to “Genome-wide association study of arabinoxylan content from а 562 hexaploid wheat collection”.

Abstract

Line 14: There are two words „using” in a sentence. Replace one of them with “with”.

Response:

Edited to 'Using GWAS with BLINK'.

Lines 19-20: “In addition, the AX level was approximately 10% higher in 3 adenine combinations than 2 guanine, 1 adenine, and 3 guanine combinations in genotypes of three genes.” Please write the names of these genes either here or in the last sentence of the abstract (Line 23).

Response:

We wrote the names of three genes in lines 19-20.

Combine in one sentence “The adenine allele of AX-95086356 exhibited a high correlation with AX levels following classification by country” (Lines 20-21) and “and the AX-95086356 SNP has 97.46% adenine alleles” (Lines 22-23), e.g. “The adenine allele, present in 97.46% in AX-95086356 SNP, exhibited a high correlation with AX levels following classification by country.”

Response:

Following your advice, I've combined them into one sentence.

Line 22-23: Please, write the genotype name after “the East Asian wheat genotype” and remove “and the AX-95086356 SNP has 97.46% adenine alleles.”

Response:

We have included the genotype names in the supplementary file 1 and removed “and the AX-95086356 SNP has 97.46% adenine alleles.”

In addition, we corrected ‘genotype contains’ to ‘genotypes contain’.

Introduction

Lines 33-34: “Some xylose residues are substituted by methyl-glucuronic acid and arabinose units, and arabinose can bind ferulate (FA) residues [6]”. What does this mean in practice? Do such substitutions bring some different properties to DF and how it is relevant to this study? Please add in a very few words what this practically implies in the context of your study.

Response:

Added the following:

“FA is the most abundant hydroxycinnamic acid derivative in plant cell walls [7], and it confers antioxidant properties to AX [8].”

Line 36: “Arabinoxylan is present in the endosperm of bran, aleurone, and bran cell wall”. This sentence is not very clear. It may imply that there is “the endosperm of bran”. (Bran, endosperm and germ are three different distinctive parts of wheat grain). This sentence needs to be unambiguous and precise. It may be also useful to briefly and precisely remand the readers of the parts of the grain, i.e. in which part of the grain aleurone and pericarp are located, and keep consistent in the text.

Response:

Modified as below:

Wheat grain comprises three main components: the bran, germ, and endosperm. The outer layers are part of the bran, which is further subdivided into three distinct layers—the testa, aleurone, and pericarp [12].

Line 38-39: “The AX of the aleurone layer and pericarp contains higher ferulic acid levels than that of starchy endosperm [12, 13].” Similar to my previous comment in Lines 33-34, the Introduction part should introduce the readers to the meaning and importance of the research, so please add very briefly to this sentence why higher ferulic acid levels are important (e.g. antioxidant properties, etc.).

Response:

Added to Lines 33-34.

Line 66-72: Replace the “Therefore, we performed a” with “Therefore, the aim of our study was to performed a” and rewrite the rest of the paragraph in such style. This should not be the part where you present your results (there is a Results section for that purpose), but the emphasis what is the aim of your study. The last sentence where the authors express their believes should be move to the conclusion.

Response:

Thanks for your advice.

We replaced the sentence “Therefore, we performed a” with “Therefore, the aim of our study was to performed a”.

Results

Lines 75-76: “Recently, we constructed a core collection of wheat resources from 614 out of 1,967 wheat lines [29].” It is not clear whether or not this core collection or a part of it is used in this study. If this material is used in this study, this sentence should be moved to the Material and methods section and elaborated. If not, the authors could compare and discuss their results with the finding in the reference no. 29 in the Discussion section.

Response:

This core collection was used in this study. Thereby this sentence has been moved to the Materials and methods section and further described.

Line 76: “A total of 567 wheat genotypes were obtained using the Axiom 35 K iSelect SNP array”. Were 567 or 562 genotypes analyzed? 

Response:

562 genotypes were analyzed. Corrected.

Line 81: How 567 (the sum of 184, 68, 72, 102, 38, 21, 77, and 5) genotypes were analyzed and clustered when it is stated in the Material and Methods that 5 out of 567 genotypes were excluded from genotyping?

Response:

562 genotypes were analyzed. Corrected.

Line 85: The “main contributions” to what? This is not precise. Did you mean that 38% and 30% of all wheat genotypes were from in Asia and America, respectively? This has to be written more clearly.

Response:

Corrected to “Asian and American wheat genotypes accounted for 38% and 30% of the wheat core collection, respectively.”

Line 91: What does this sentence mean? “Groups I and III genetic traits were derived from various countries.” Please, rewrite it and be clearer.

Response:

We rewrite it as “Groups I and III were a mix of genotypes from various countries.”

Line 92: This sentence should be in MM section.

Response:

This sentence has been deleted because it has the same content in the MM section.

Lines 93-94: This statement has no meaning. The author should describe how PCA divided/grouped/ classified the genotypes.  

Response:

We described how PCA segmented/grouped/classified genotypes.

“To better demonstrate the population structure of our study, we performed principal component analysis (PCA) using the 562 wheat genotypes collected from 46 countries. When the genotypes were grouped by PCA based on continent, PC1 of the PCA biplot showed clear clusters of the Asian and American continental genotypes; however, the genotypes of four continental sets (Africa, Eurasia, Europe, and Oceania) were mostly mixed in the middle with no clear clusters”

Lines 100-101: The Results section should contain only the results of this study and should not have any citations. This sentence belongs to MM.  

Response:

I deleted this sentence.

I would like to see the full ANOVA table of the AX with df, p values and levels of significance for the genotypes and countries, as the source of variation.

Response:

We have included this information in supplementary file 1.

Figure 2a. Replace “germplasm” with “genotypes”.

Response:

Replaced.

Figure 2b. It would be useful to see the letters above the bars to see whether or not there are significant differences among the average values of AX in different countries. Why only 11 countries are presented in the figure? From how many countries were the genotypes anyway? The authors discussed (Lines 302-304) that some groups were underrepresented (with few accessions) and thus not presented, but this should be also  appropriately stated in the results, just before showing the results, so readers can follow.   

Response:

Thanks for your comments.

We marked the significant differences between AX mean values for different countries with letters above the bars. Genotypes were collected from 46 countries, and AX content was compared by selecting countries with more than 10 genotypes (France, which lacks resources, was excluded from the existing data.). A description has been added to the text, and related information has been added to Supplementary Table 1 and Supplementary File 1.

Lines 113-115: This is a part of the Material and methods section. It should be move there.

Response:

This sentence has been moved to the Materials and Methods section.

Lines 136-137, 138: Cite the databases in MM. The same is for [34].

Response:

This sentence has been moved to the Materials and Methods section.

Figure 6 Word “East” is missing on a graph.  

Response:

Corrected.

Figure 8a. Replace “germplasm” with “genotypes”.

Response:

Replaced.

Discussion

Discussion is rather short and should be elaborated a little bit more. For example, what analysed genotypes could be potentially valuable as a source of AX for wheat improvement. Discuss the difference between the level of AX in genotypes from South Korea (and e.g. Japan) and from other countries. How could be explained. What are the previous findings on AX content?

Response:

As advised, the effect of environment and genotype on arabinoxylan content was further discussed in the discussion section.

Material and methods

Lines 346-347: “The resources analyzed in this study were a subset of the resources used by Yang et al. (2021) and 567 hexaploid wheat varieties collected from 47 countries [31]”. Having stated 567 hexaploid wheat varieties in the second part of the sentence, what did the authors mean by “a subset of the resources”? The statement suggests that there is a subset of some resources AND 567 varieties.

Response:

Yang et al (2021) analysed the arabinoxylan content using 614 wheat genetic resources. Of those, 562 resources were analysed in GWAS. We changed 'were a subset of the resources' to ‘included some of the genotypes’

The cited paper [31] with the information about the varieties and AX context that the authors used for this study is in Korean language. Please, add the remaining genotypes, the countries and their AX values to the 41 varieties in the supplementary file. All 562 genotypes with their data must be included in the supplementary file.

Response:

We have included all 562 genotypes and information such as arabinoxylan content in Supplementary Table 1 and Supplementary File 1.

Lines 89, 102, 104, 109, 164, 172, 174, 177, 180, 181, 183, 196, 197, 203, 234, 274, 294, 235, 346, 359 (and probably some more): The term “resources” in this context is a bit odd and inappropriate. Please, use more suitable terms, such as genotypes, accessions or varieties.

Response:

We changed resources to genotypes.

Lines 358-359: If out of 567 accessions, 562 were genotypes and 5 were excluded from the analyses due to abnormal phenotype, it cannot be claimed that all 567 accessions were analyzed. It is 562 then. This must be corrected in the whole manuscript.

Response:

562 genotypes were analyzed. 567 was changed to 562.

Conclusion

Lines 388: Correct “three SNPs than in other countries” to “three SNPs than in wheat accessions from other countries”. The wheat varieties can be in/from other countries, not the SNPs. 

Response:

Thanks for your comment. Corrected.

Reviewer 3 Report

I believe that the article is written at the proper level of modern scientific research. I do not have any critical remarks on the essence of setting up the experiment or on its interpretation. The authors should eliminate several inaccuracies and typos found in the text. Conduct more thorough text editing

Author Response

Reviewer 3

Comments and Suggestions for Authors

I believe that the article is written at the proper level of modern scientific research. I do not have any critical remarks on the essence of setting up the experiment or on its interpretation. The authors should eliminate several inaccuracies and typos found in the text. Conduct more thorough text editing

Response:

Thanks for your comments.

Removed incorrect sentences and typos. And I've included additional information.

Round 2

Reviewer 1 Report

In the revised version, the authors complied with my minor remarks (no. 3 and 4). Accordingly, they specified the number of studied accessions and corrected the names of the nucleotides.

I also suggested that authors calculate the LD in the studied population (remark no.1), extend the search for candidate genes based on LD, and explain the putative relationship of the identified candidate genes (CGs) with the arabinoxylan level. In this version, the authors present analysis of LD between the three selected significant SNPs, but have not used this knowledge to search for other promising candidate genes. Neither they explain the relationship between the identified CGs and the trait under investigation.

However, they have not taken into account my major remark:

The authors have built the manuscript and the conclusions based on differences between the allelic variants at eight significant SNP loci, or allelic combinations at three selected SNP loci. This is illustrated on figures 5, 6, 7b and 8d,e. However, as I wrote in my previous report, the authors did not mark what the bars on the columns stay for. I suppose, the bars denote the standard deviations. Then, looking at the bars/standard deviations on these figures, the vast majority of differences between the allelic variants are NOT significant: Fig. 5 – no significant differences at all; Fig. 6 - only the difference between the alleles at locus AX-95086356 for genotypes from East Asia is significant; Fig. 7 - no significant differences at all.

In addition, on Fig. 2b, only the differences between the arabinoxylan contents in genotype sets from Canada and Japan, and Canada and South Korea are significant. The rest differences are NOT significant.

I do not understand why the authors have put asterisks and letters above the columns on the figures claiming these differences are significant.

I do not agree with the authors response “The three alleles of AX-5086356, AX-94470319 and AX-94713015 showed statistically significant differences in Fig 5.” Looking at Fig. 5, I do not see any significant difference between the allelic variants at all eight SNP loci.

Regarding Fig. 6, I do not agree with the authors response “In addition, AX content did not show a significant difference in these three alleles in the United States and Mexico, but showed a significant difference in East Asia and Korea”.  The authors added new ANOVA tables as supplementary material. According to Supplementary file 1, sheets 6 to 9 marked as E.Asia, S. Korea, Mexico and USA, respectively, the following differences between the allelic variants are statistically significant: E. Asia – at all three SNP loci; S. Korea – AX-94470319 and AX-94713015; Mexico – AX-95086356. What I see on Fig. 6 is that there is significant difference only between the alleles at locus AX-95086356 for genotypes from East Asia. There is obviously a discrepancy between the presented figures, the additional ANOVA results, and the authors’ explanations in response to my remarks.

Regarding Fig. 7, I do not agree with the authors response “And, as shown in Fig 7, since the three adenine combinations showed a statistically higher AX content than the three guanine combinations, these results are considered to be significant alleles for the arabinoxylan content“.  In the Supplementary file 1, sheet 5 marked as “3 alleles combinations”, the provided ANOVA table shows that generally the allelic combinations affect the AX content. But pairwise comparisons are missing. It is possible that the 3AA and 3GG genotypes differ, but the reader should be convinced by relevant information, which is missing. In fact, when I look on Fig. 7, I see no differences, while authors claim that there are significant differences  between 3AA and 2AA1GG, between 3AA and 1AA2GG, and between 3AA and 3GG; also between 2AA1GG and 1AA2GG and between 2AA1GG and 3GG; the letter-designations over the columns are (from left to right) a, b, c, c. Assuming that pairwise comparisons confirm the significant difference between 3AA and 3GG (as the authors now claim in their response), and insignificant differences among the rest combinations, than the letters order should be (from left to right) a, ab, ab, b.

Also, both Fig. 6b (results for the South Korea) and Fig. 8d   compare the allelic variants AA and GG for the AX content in South Korean set of accessions. However, while on Fig. 6b, the two allelic variants are marked as significantly different for loci AX 94470319 and AX-94713015 (which is not correct), on Fig. 8d these differences are marked as insignificant (which is correct).

I do not understand the meaning of the added text in the Discussion section (lines 354-360).

I do not understand the meaning of the following sentence (lines 364-365): “The SNPs found in this study have limitations in the selection of varieties with a high AX content among varieties with high AX.

Author Response

In the revised version, the authors complied with my minor remarks (no. 3 and 4). Accordingly, they specified the number of studied accessions and corrected the names of the nucleotides.

I also suggested that authors calculate the LD in the studied population (remark no.1), extend the search for candidate genes based on LD, and explain the putative relationship of the identified candidate genes (CGs) with the arabinoxylan level. In this version, the authors present analysis of LD between the three selected significant SNPs, but have not used this knowledge to search for other promising candidate genes. Neither they explain the relationship between the identified CGs and the trait under investigation.

Response:

LD analysis was performed on the remaining 5 SNPs (Supplementary Figure S1). And the description of LD was moved to results section 2.4 (Candidate gene identification) and explained in connection with the candidate genes (line 142-150)

However, they have not taken into account my major remark:

The authors have built the manuscript and the conclusions based on differences between the allelic variants at eight significant SNP loci, or allelic combinations at three selected SNP loci. This is illustrated on figures 5, 6, 7b and 8d,e. However, as I wrote in my previous report, the authors did not mark what the bars on the columns stay for. I suppose, the bars denote the standard deviations. Then, looking at the bars/standard deviations on these figures, the vast majority of differences between the allelic variants are NOT significant: Fig. 5 – no significant differences at all; Fig. 6 - only the difference between the alleles at locus AX-95086356 for genotypes from East Asia is significant; Fig. 7 - no significant differences at all.

Response:

Thanks for your comments.

Bars represent standard deviations, of course.

As you pointed out, the bar graph did not seem appropriate to show the difference in AX content, so we changed it to a box graph instead of a bar graph to explain the difference in AX content. In addition, instead of the expression 'significant', the average value of AX content was compared and explained, and the manuscript was generally revised.

The contents of Figure 6 have been amended (section 2.6).

An explanation was added comparing average AX values as percentages.

As you point out, AX content was not significant in the Mexico and the USA for the AX-94470319 and AX-94713015. We propose that in AX-94470319 and AX-94713015, differences by genotype are difficult to see because the AX content of the Mexico and USA genotypes is lower than the overall average. Added this comment to discussion section.

 In addition, typo was corrected (% à mg/g).

Figure 7 was also expressed as a box graph, and the AX content was compared in percentage (section 2.7). “AX content was higher in all three adenine genotypes (29.37%) and in the two adenine genotypes (18.52%) than in all guanine genotype.”

In addition, on Fig. 2b, only the differences between the arabinoxylan contents in genotype sets from Canada and Japan, and Canada and South Korea are significant. The rest differences are NOT significant.

Response:

We also expressed Figure 2 as a box graph.

In figure 2, Japan and South Korea genotypes showed average AX content was 54.51 mg/g and 56.91 mg/g, respectively. In contrast, genotypes from China, Afghanistan, and Canada had AX contents of 35.88 mg/g, 41.29 mg/g, and 42.11 mg/g, respectively. And genotypes from Ethiopia, Mexico, Russia, Turkey, and USA had AX contents of 45.75 mg/g, 48.56 mg/g, 50.03 mg/g, 44.95 mg/g, and 47.49 mg/g, respectively. There is a 10-20% difference in average ax content compared to Korean and Japanese resources. However, there was high variability in the Afghanistan, China, and Mexico genotypes (Figure 2). Added this to the Results section.

I do not understand why the authors have put asterisks and letters above the columns on the figures claiming these differences are significant.

Response:

As responded above, the bar graph did not seem appropriate to show the difference AX content, so we changed it to a box graph instead of a bar graph to explain the difference in AX content. In addition, instead of the expression 'significant', the average value of AX content was compared and explained, and the manuscript was generally revised.

I do not agree with the authors response “The three alleles of AX-5086356, AX-94470319 and AX-94713015 showed statistically significant differences in Fig 5.” Looking at Fig. 5, I do not see any significant difference between the allelic variants at all eight SNP loci.

Response:

The bar graph of figure 5 was expressed as a box graph. The description has been amended as follows. “The genotypes containing the SNP adenine genotype had an average AX grain content of approximately 5–9 mg/g higher than that containing guanine in three of the eight candidate SNPs: AX-95086356 (19.71%), AX-94470319 (17.71%), and AX-94713015 (10.48%).”

And a description of AX contents for the remaining SNPs was also included.

In addition, we corrected the graph in figure 5g because the genotype of AX-94502724 has a heterozygous (CT) and thymine genotype in Supplementary Table S1 and does not include a cytosine genotype.

Regarding Fig. 6, I do not agree with the authors response “In addition, AX content did not show a significant difference in these three alleles in the United States and Mexico, but showed a significant difference in East Asia and Korea”.  The authors added new ANOVA tables as supplementary material. According to Supplementary file 1, sheets 6 to 9 marked as E.Asia, S. Korea, Mexico and USA, respectively, the following differences between the allelic variants are statistically significant: E. Asia – at all three SNP loci; S. Korea – AX-94470319 and AX-94713015; Mexico – AX-95086356. What I see on Fig. 6 is that there is significant difference only between the alleles at locus AX-95086356 for genotypes from East Asia. There is obviously a discrepancy between the presented figures, the additional ANOVA results, and the authors’ explanations in response to my remarks.

Response:

Thanks for the advice.

As explained above, the AX contents of Figure 6 have been amended.

As you point out, AX content was not significant in the Mexico and the USA for the AX-94470319 and AX-94713015. An explanation was added comparing average AX values as percentages. So, we propose that in AX-94470319 and AX-94713015, differences by genotype are difficult to see because the AX content of the Mexico and USA genotypes is lower than the overall average. Added this comment to discussion section.

Regarding Fig. 7, I do not agree with the authors response “And, as shown in Fig 7, since the three adenine combinations showed a statistically higher AX content than the three guanine combinations, these results are considered to be significant alleles for the arabinoxylan content“.  In the”, the provided ANOVA table shows that generally the allelic combinations affect the AX con Supplementary file 1, sheet 5 marked as “3 alleles combinations tent. But pairwise comparisons are missing. It is possible that the 3AA and 3GG genotypes differ, but the reader should be convinced by relevant information, which is missing. In fact, when I look on Fig. 7, I see no differences, while authors claim that there are significant differences  between 3AA and 2AA1GG, between 3AA and 1AA2GG, and between 3AA and 3GG; also between 2AA1GG and 1AA2GG and between 2AA1GG and 3GG; the letter-designations over the columns are (from left to right) a, b, c, c. Assuming that pairwise comparisons confirm the significant difference between 3AA and 3GG (as the authors now claim in their response), and insignificant differences among the rest combinations, than the letters order should be (from left to right) a, ab, ab, b.

Response:

Figure 7 was also expressed as a box graph, and the AX content was compared in percentage (section 2.7). Instead of the 'significant' expression, the average was compared and explained. So ANOVA results were deleted.

AX content was higher in three adenine genotypes (29.37%) and two adenine genotypes (18.52%) than in all guanine genotypes.

Also, both Fig. 6b (results for the South Korea) and Fig. 8d   compare the allelic variants AA and GG for the AX content in South Korean set of accessions. However, while on Fig. 6b, the two allelic variants are marked as significantly different for loci AX 94470319 and AX-94713015 (which is not correct), on Fig. 8d these differences are marked as insignificant (which is correct).

Response:

Thanks for your comments.

The Korean genotypes in Figure 6 and the Korean genotypes in Figure 8 are different. Among these genotypes, there is only one overlapping resource (Keumkang), and the remaining 40 genotypes do not overlap with a 562 wheat collection. The genotypes in Figure 8 are the 41 varieties we have currently bred and selected. This is an independent analysis. Information on 41 varieties of Korean resources is indicated in supplementary table S4, and information on 127 Korean varieties in the wheat collection is indicated in supplementary file 1.

I think it's confusing because I didn't explain this in detail. Added explanation in result section 2.8.

The resources in figure 8 do not reflect the diversity of Korean genotype. And overall, the AX content was high, and there was no significant difference by genotype in the three SNPs.

It was proposed that it was difficult to distinguish the difference in AX content by genotypes by candidate SNP between genotypes with generally low or generally high AX content, and was included in the discussion section.

I do not understand the meaning of the added text in the Discussion section (lines 354-360).

I do not understand the meaning of the following sentence (lines 364-365): “The SNPs found in this study have limitations in the selection of varieties with a high AX content among varieties with high AX.”

Response:

We deleted that sentence. Instead, the following has been added to the discussion section.

“Three SNP genotypes (AX-95086356, AX-94470319, and AX-94713015) had an average AX grain content that was approximately 5–9 mg/g higher in the adenine genotype than in the guanine genotype. Among these, the average AX content was higher in the adenine genotype than in the guanine genotype of AX-95086356 from Mexico, the USA, and East Asia. However, the average AX content was similar in adenine and guanine alleles in Mexico and the USA; there were differences only in East Asia and South Korea in both AX-94470319 and AX-94713015 SNPs. We propose that these results are because of the low average AX content in Mexico and the USA. The AX content is variable, and low on average from the 31 Chinese genotypes. The estimated average increase in AX content in East Asia is owing to the relatively large number of genotypes from South Korean genotypes (127).

             The average AX content in the three adenine genotype combinations was slightly higher than in each genotype and higher than in the three guanine genotype combinations. However, there was no difference in AX content among the 41 domestic varieties according to SNPs. This is probably owing to the selection and breeding of genetically similar cultivars, and the average AX content of 41 Korean cultivars is estimated to be similarly high. These results suggest that the SNPs found in this study have limitations in the selection of varieties with a high AX content among genotypes with a high or low AX content.”
